# New Insights for Red Propolis of Alagoas—Chemical Constituents, Topical Membrane Formulations and Their Physicochemical and Biological Properties

**DOI:** 10.3390/molecules25245811

**Published:** 2020-12-09

**Authors:** Valdemir C. Silva, Abiane M. G. S. Silva, Jacqueline A. D. Basílio, Jadriane A. Xavier, Ticiano G. do Nascimento, Rose M. Z. G. Naal, Maria Perpetua del Lama, Laila A. D. Leonelo, Naianny L. O. N. Mergulhão, Fernanda C. A. Maranhão, Denise M. W. Silva, Robert Owen, Ilza F. B. Duarte, Laisa C. G. Bulhões, Irinaldo D. Basílio, Marília O. F. Goulart

**Affiliations:** 1Laboratory of Pharmaceutical Technology, Research Program Post-Graduation in Pharmaceutical Sciences, Institute of Pharmaceutical Sciences, Federal University of Alagoas (UFAL), Maceió, Alagoas 57072-970, Brazil; valldemir_costa@hotmail.com (V.C.S.); abiane_gomes@hotmail.com (A.M.G.S.S.); ticianogn@yahoo.com.br (T.G.d.N.); naiannylivia@hotmail.com (N.L.O.N.M.); ilzafernandabd@hotmail.com (I.F.B.D.); laisacarolina@hotmail.com (L.C.G.B.); 2Institute of Chemistry and Biotechnology, Federal University of Alagoas, Maceió, Alagoas 57072-970, Brazil; jacarantesdiniz@hotmail.com (J.A.D.B.); jadrianexavier@iqb.ufal.br (J.A.X.); 3Department of BioMolecular Sciences, Faculty of Pharmaceutical Sciences of Ribeirão Preto. Avenida do Café, s/n, Ribeirão Preto, São Paulo 14040-903, Brazil; rosenaal@usp.br (R.M.Z.G.N.); mpemdel@fcfrp.usp.br (M.P.d.L.); lailadeliberto@yahoo.com.br (L.A.D.L.); 4Institute of Biological Science and Health, Federal University of Alagoas, Maceió, Alagoas 57072-970, Brazil; fernanda.maranhao@icbs.ufal.br (F.C.A.M.); denise.wanderlei@gmail.com (D.M.W.S.); 5Division of Preventive Oncology, German Cancer Research Center, Im Neuenheimer Feld 460, 69120 Heidelberg, Germany; r.owen19sheep52@gmail.com

**Keywords:** phytochemical screening, chromatographic profile, sodium carboxymethylcellulose, biopolymer, allergenic activity, anti-staphylococcal, biopolymer

## Abstract

The main objectives of this study were to evaluate the chemical constitution and allergenic potential of red propolis extract (RPE). They were evaluated, using high performance liquid chromatography (HPLC) and the release of β-hexosaminidase, respectively. A plethora of biologically active polyphenols and the absence of allergic responses were evinced. RPE inhibited the release of β-hexosaminidase, suggesting that the extract does not stimulate allergic responses. Additionally, the physicochemical properties and antibacterial activity of hydrogel membranes loaded with RPE were analyzed. Bio-polymeric hydrogel membranes (M) were obtained using 5% carboxymethylcellulose (M1 and M2), 1.0% of citric acid (M3) and 10% RPE (for all). Their characterization was performed using thermal analysis, Fourier transform infrared (FTIR), total phenolic content, phenol release test and, antioxidant activity through 2,2-diphenyl-1-picrylhydrazyl radical (DPPH) and Ferric Reducing Antioxidant Power (FRAP). The latter appointed to the similar antioxidant capacity of the M1, M2 and M3. The degradation profiles showed higher thermostability to M3, followed by M2 and M1. The incorporation of RPE into the matrices and the crosslinking of M3 were evinced by FTIR. There were differences in the release of phenolic compounds, with a higher release related to M1 and lower in the strongly crosslinked M3. The degradation profiles showed higher thermostability to M3, followed by M2 and M1. The antibacterial activity of the membranes was determined using the disc diffusion assay, in comparison with controls, obtained in the same way, without RPE. The membranes elicited antibacterial activity against *Staphylococcus aureus* and *Staphylococcus epidermidis*, with superior performance over M3. The hydrogel membranes loaded with RPE promote a physical barrier against bacterial skin infections and may be applied in the wound healing process.

## 1. Introduction

Hydrogels consist of a polymeric three-dimensional chain with considerable water and biological fluid absorption capacity. They are used as dressings for wounds and burns, due to their porous and hydrophilic characteristics. They also ensure flexibility and selective diffusion to the metabolites of surrounding tissues and oxygen, important factors for the recovery of injured tissues [1].

Polymeric matrices based on sodium carboxymethylcellulose (NaCMC) form transparent biofilms with high mechanical resistance and, healing potential when in contact with the skin. They promote a moist environment that stimulates collagen synthesis, accelerates the growth of epithelial cells and promotes the debridement of necrotic tissue [2]. To enhance the therapeutic properties, hydrogels can serve as matrices for synthetic or natural bioactive compounds that accelerate the treatment. Natural products, used in traditional medicine, are among the options for incorporation in polymeric matrices, providing the synergistic effect to accelerate the wound healing process. In this scenario, red propolis appears as an alternative for incorporation in polymeric matrices, due to its biological properties, cost-effectiveness and low adverse effect [3].

The propolis is a natural product complex formed by a mixture of wax, pollen, salivary secretions, resinous, gummy and balsamic substances collected from various plant sources and biotransformed by different species of bees [4]. In Brazil, there are 13 types of propolis and their classification is made according to geographic location, botanical origin, color and physicochemical proprieties [5]. The red propolis was classified as the 13th group and can be found, mainly, in the Brazilian coastal regions of states of Alagoas, Sergipe, Pernambuco, Bahia and Roraima [5,6,7]. These Brazilian states have particularities regarding climatic seasonality, plant variability and soil composition. These distinct characteristics, such as geographical location, flora and endaphoclimatic conditions, strongly influence the chemical composition and their biological activities [7,8].

Since their discovery, the red propolis has been severely investigated for their chemical and biological properties [5]. A species of the Leguminosae family (*Dalbergia ecastophyllum*) is the main botanical source of red propolis found in Brazil. Their constituents are different from other propolis varieties and are important as markers for their identification, such as formononetin, isoliquiritigenin, medicarpine, biochanin A and pinocembrin [9].

The propolis is known for its extensive biological activities related to skin healing [10,11,12,13,14], as well as an antioxidant [15,16], antibacterial [8,9], antifungal [11] and anti-inflammatory activities [14]. These activities are attributed to phenolic compounds and are considered important for the treatment of wounds. It also has the antiallergic potential [17], evidenced by the deregulation of type I allergy, this fact is related to mast cell degranulation and the production of a chemical mediator of the allergic response [17,18]. This property is especially relevant for the applications of natural compounds in the development of biomaterials that promote wound healing. Bee products are used in the production of cosmetics and the development of drugs and can cause allergic reactions [19]. Dermatitis and contact allergy problems have also been reported in vitro [20] and for beekeepers and propolis consumers [21], which emphasizes the importance of assessing the allergenic properties of hydroalcoholic extracts of red propolis (RPE) [22,23,24], on account of their chemical composition.

Therefore, the present study was conducted to characterize the phenolic composition of RPE and the development, physicochemical characterization, analysis of the antibacterial activity and allergenic potential of the bio-polymeric hydrogel membranes (M), based on NaCMC, loaded with RPE (M1, M2 and M3), in comparison to the controls P1, P2 and P3, prepared in the same manner, without RPE. 

## 2. Results

### 2.1. Determination of the RPE Chemical Markers

The major phenolic compounds in the RPE were identified and confirmed by LC-Orbitrap-FTMS and are shown in Figure 1 and listed in Table 1.

The analyses revealed a diverse range of natural products, belonging to several groups. Among the detected phenols and polyphenols, there were phenolic acids, flavan-3-ols (catechins), flavonols, chalcones, isoflavones, isoflavans, pterocarpanes and biflavonoids, which eluted between 3 to 25 min. A further group of substances with characteristic molecular weights of a terpene (β-amyrin) and guttiferones (guttiferones E and C) were identified at 39.24 min.

Compounds, such as retusapurpurin A and B, xanthokimol, vestitol and neo-vestiol, in addition to new chalcones, were also identified in the same elution range. Formononetin, liquiritigenin, daidzein, isoliquiritigenin and biochanin A were identified by LC-Orbitrap-FTMS in the red propolis from Bahia, Alagoas, Paraíba and Sergipe [16,25]. The results presented here corroborate the identity of Brazilian red propolis [7].

### 2.2. Membrane (M) Production Loaded with Red Propolis Extract

Table 2 summarizes the components present in the membrane compositions. There was little difference in the visual appearance between M2 and M3. M1 was more homogeneous than the other membranes, which gave a more evident visualization of the coloration of the red propolis. Membranes presented significant flexibility, homogeneity and resistance to handling, varying according to the formulation (M1 > M2 > M3, respectively). On the other hand, the flexibility of M3 was similar to that of the M2.

The addition of a wetting agent to the formulation of M1 may have influenced the achievement of the best macroscopic characteristics, making it more stable and homogeneous in relation the other formulations. Wetting agents assist in hydration and prevent loss of water from the formulations. However, no studies to date have confirmed the relationship of these components to the stability of the rheological parameters of the formulation, despite the intuitive relationship between the viscosity parameter and the addition of humectants [26].

According to the results of the present study, M3, although being developed using a higher concentration of the crosslinking agent, in comparison to the other membranes, showed similar flexibility to M2 and lower than M1. There is an ideal amount of citric acid required to develop the flexibility and resistance characteristics to handling the membranes. High concentrations may lead to excess crosslinks, limiting the mobility of the substances [27].

### 2.3. Thermogravimetric Analysis (TGA)

To compare the thermal degradation of the formulations developed with RPE, thermogravimetric analyses of the three membranes (M1, M2, M3) loaded with RPE were used, in comparison with P1, P2 e P3 (controls without RPE). Figure 2A,B represent the thermal profiles of the membranes along with RPE. The presence of two degradation events for the RPE was observed: (i) a mass loss of 10.2% associated with the removal of water adsorbed in the extract, in the temperature range 91.8 °C to 141.4 °C. (ii) there was also a marked mass loss (76.5%) occurring between 283.3 °C and 371.9 °C and a total loss of 86.8%. It is believed that in this temperature range, degradation of the phenolic compounds occurs by thermo-oxidation processes. Recent studies have also demonstrated this thermo-oxidative behavior at temperatures above 270 °C [28]. 

The thermograms of the hydrogel-based polymer membranes are shown in Figure 2A,B with multiple degradation stages. M1 presented a different thermal profile compared to M2 and M3. M1 presented four thermal events while in M2 and M3, five degradation events with similar characteristics were observed. The first event for all membranes revealed a mass loss of 10.6% (M1), 6.5% (M2) and 7.1% (M3), corresponding to the water adsorbed into the polymer matrix through the hydrolysis and causing the loss of -OH groups by dehydration, at different temperatures. FT-IR spectra showed that the bands in the region of 3384–3466 cm^−1^ (relative to νOH) was discretely more intense in the M1 and corroborates with the findings of the thermogravimetric analysis of the first event with higher moisture loss when compared to M2 and M3 (Figure 2A vs. Figure 3).

In the second and third thermal events of the degradation of the membrane, it was possible to observe, according to the derivative thermogravimetric analysis (DTG) and thermograms (TG) of the placebo membranes (Figure 2C,D) that there was temperature differences in the degradation stages and the proportion of mass losses when compared with those loaded with RPE (Figure 2A,B). The second event of the P1 occurs at degradation temperatures similar to that of citric acid (150–200 °C) and may be related to low crosslinking efficiency due to the formation of weak esters with OH groups of NaCMC, which can be broken with increasing temperature [29]. On the other hand, it was observed that the degradation profile of M1 was different in this second event of degradation, which occurred at a higher temperature, between 259.9–323.8 °C, with a mass loss of 51.8%. The higher stability identified in M1 can be attributed to the higher degree of crosslinking formed by esters of citric acid with OH groups of phenolic compounds and (NaCMC), thus reducing the mobility of the polymer chains. The second event of M2 and M3 shows thermal degradation between the temperature range of 228.1–269.7 °C and 222.7–266.4 °C and mass losses of 28.6% and 18.5%, respectively. These membranes form more stable chains when compared to P1 and P2 and P3 (Figure 2B vs. Figure 2D) This fact can be related directly to the absence of the wetting agent and to the addition of RPE, allowing a better formation of crosslinks and consequently more stable polymerization. A double amount of the crosslinking agent provided higher thermal stability for M3.

The third stage of degradation shows the decomposition of polymer chains of NaCMC and RPE. M2 and M3 presented a similar temperature zone of 304.6–336.7 °C and 285.4–336.2 °C with weight losses of 20.7% and 31%, respectively. The thermal decomposition of NaCMC involves the decarboxylation process, removing the C=O group of the polysaccharide between the temperature range of 250 to 400 °C [1]. Part of the decarboxylation of NaCMC and M1 occurred in the second stage of degradation, ending in the third event with a mass loss of 5.7%.

M2 and M3 presented an additional fifth thermal event when compared to M1. The latter stages of thermal decomposition of the membranes are related to the degradation of chemical clusters that require greater thermal energy to be broken and charred. The total mass losses of M1, M2 and M3 correspond to 73.4%, 73.7% and 75.8% respectively. DTG curves of the membranes loaded with RPE and RPE alone can be observed in Figure 2B. M1, M2 and M3 demonstrated maximum degradation temperatures of 292.6, 255.7 and 303.5 °C respectively, whereas for the RPE it was 320.2 °C. Therefore, it is suggested that M3 presents greater thermal stability due to the double amount of citric acid used as a crosslinking agent.

### 2.4. ATR Coupled to Fourier Transform Infrared (ATR-FTIR)

The main functional groups of the membranes were investigated by Attenuated Total Reflectance-FourierTransform Infrared Spectroscopy (ATR-FTIR). Figure 3 shows the ATR-FTIR spectra of the RPE and the RPE-incorporated membranes. For the RPE, the stretching of the bands OH at 3372 cm^−1^ representing phenolic hydroxyl group was identified. The bands C-H_3_ (sp^3^) at 2932 cm^−1^ and, C-H_2_ (sp^3^) at 2856 cm^−1^ are peaks from the residual wax present in the hydroalcoholic extract. Bands at 1618, 1508 and 1448 cm^−1^ correspond to the *ν*C=C of the aromatic ring and at 1030 cm^−1^ to the C-O bond (for flavonoids), as well as the band at 836 cm^−1^, relative to an angular deformation outside the aromatic C-H plane, according to the literature [28,30].

M1, M2 and M3 showed a similar profile in comparison with the RPE, proving the incorporation of the extract into the polymer matrix. The FTIR spectra of membranes revealed the simultaneous presence of sodium carboxylate (-COONa) and carboxylic acid (-COOH) in the NaCMC hydrogel, attributed respectively to bands in the region between 1622–1613 cm^−1^ and 1735–1729 cm^−1^ [31]. The reaction between citric acid and the hydroxyl groups of the NaCMC leads to crosslinkings on the polymer matrix [32]. The crosslinking step through acidification of the hydrogel with citric acid (CA) caused a change in the spectrum, suggesting the substitution of Na^+^ for H^+^ in the chains of the NaCMC polymer. The reduction of the intensity of the -OH bands (3384 to 3466 cm^−1^) and the increase of the peak intensity in the region between 1735 to 1729 cm^−1^ of citric acid-crosslinked membranes were observed. It is suggested that the increase of citric acid concentration can cause the consumption of -OH due to intramolecular and intermolecular hydrogen bonds during the crosslinking reaction [33]. This consumption was observed mainly in the membranes without humectant (propylene glycol) and shown in M3, containing a double concentration of citric acid. This shows an increasing degree of crosslinking with a higher percentage of citric acid crosslinker added to the membrane. Depending on the increase in the concentration of CA, a decrease in the band representing sodium carboxylate (-COONa) and an increase in the band of carboxylic acid was observed (-COOH) [31]. 

### 2.5. Total Phenol Content (TPC) as Release Test

The pH values of the media affect NaCMC membranes, due to the presence of carboxylic groups. In the pH range, 6.4–7.4, swelling is observed. When the membranes are immersed in buffered saline, carboxyl (-COOH) groups ionize, eliciting repulsion on the molecule, which increases the swelling rate of the hydrogel [34]. Probably because of this capacity of expansion, the incorporation of the RPE into the formulation led to the entrapment of the propolis chemical components. The retention percentages of the phenolic content were obtained for the three formulations, as described in Table 3.

The amounts of RPE used were equal for the 3 membranes. M2 and M3 presented similar percentages of total phenol content and for M1, it was significantly higher. The polyphenolic composition of the membranes did not compromise stability, antioxidant activity (Table 3).

Release of phenolic compounds from the membrane occurred gradually and stabilized only after 48 h (Figure 4). The M1 gave the highest percentage of release as a function of time, reaching about 63% of its total content in 24 h. The first hour of release may be related to increased M1 swelling capacity and, consequently, a higher release of phenolic compounds. This fact can be explained by the presence of propylene glycol in the membrane composition. The propylene glycol binds to the polymer chains of the hydrogel through physical crosslink bonds [32]. When the membrane gets in contact with the phosphate buffer solution and remains at a constant temperature for a given period, propylene glycol is stimulated to release the entrapped active chemical principles (RPE).

The M3, with the highest concentration of the crosslinking agent, released a concentration of phenolic compounds slightly lower than M2 (Figure 4), which may indicate a higher crosslinking efficiency, when a 1% citric acid solution was used in M3.

### 2.6. Antioxidant Activity and Total Phenol Content

The total phenol content (TPC) was higher for M1 and slightly lower for M2 and M3 (Table 3). This finding is in accordance with the release test, in which M1 showed the highest value as a function of time (Figure 4). The results obtained in the DPPH and FRAP (Table 3) assays revealed a similar antioxidant capacity for the M1, M2 and M3. 

Reports from the literature have shown variations in the capacity of RPE to inhibit the DPPH^●^ with the respective IC_50_ values, varying highly at 270 μg /mL [35], 8 μg /mL [36], 57 μg /mL [37]. The variations in this activity may be related to the variability of the chemical composition of propolis, due to the seasonal and flora changes in the collection regions and time. 

### 2.7. Antimicrobial Activity

The initial size (10 mm) of the membranes after 24 h in Müller-Hinton agar (MHA) medium was distended to approximately 14 mm (M3), 19 mm (M2) and 22.5 mm (M1); however, in the M1, it was more significant, presumably due to the absence of the crosslinker agent, citric acid (CA). Therefore, the inhibition zone was measured in mm from the edge of each membrane already distended, without considering the initial size and the distension. This distension is considered beneficial for the treatment of skin wounds.

In this study, all membranes (M) loaded with RPE elicited antibacterial activity, with inhibition zones around the membranes in most of the assays. This effect was observed with all tests with *S. aureus* (reference and clinical strains) and *S. epidermidis* (only a clinical strain), by comparing with control-membranes (P), after 24 h. Nevertheless, these features were not observed with *P. aeruginosa* strains, except for M3. The membranes showed different antibacterial performance: for example, M3 produced significantly higher inhibition zones, compared to the other membranes (M1 and M2) in Gram-positive strains such as *S. aureus*, as shown in Figure 5b,c; *S. epidermidis* in Figure 5d and *P. aeruginosa* in Figure 5f,g.

Descriptive analysis of the interaction between halo and species by the standard deviation (SD) related to each membrane showed the best inhibitory performance of both species of *Staphylococcus* on *P. aeruginosa* (*p* < 0.001) in Figure 5. M1 and M2 contributed to a reduction of their growth; however, the inhibition was higher in M3 (halo = 4.33 ± 0.58 mm). In general, the most effective composition was M3 (inhibition zones ranging from 7.00 mm to 2.67 ± 0.29 mm), which was formulated, as described (Table 4). 

These results were also supported by the *post-hoc* comparisons (corrected by Tukey *p* < 0.001). The control membranes (P1 and P2) revealed no antimicrobial activity against *S. aureus* and *P. aeruginosa*, except for P1 that produced a significant halo (*p* < 0.001) against *S. epidermidis* and effectively inhibited by P3 (halo = 3.33 ± 0.29 mm) but not by P2. The highest antibacterial activity was for M2, M1 and M3 (halo = 5.67 ± 0.29 mm), in increasing order. Otherwise, M3 promoted antibacterial activity against all bacteria, with a difference in the measurement of halo in comparison with P3 (citric acid 2×), which generated a smaller halo. Among the Gram-positive bacteria tested, the difference between the M3/P3 halos in the *S. aureus* clinical strain was lower (4.33/4.83 mm) than that observed in reference strains of *S. aureus* (7.00/1.00 mm) and *S. epidermidis* (5.67/3.33 mm).

Table 5 shows the analysis of the variance (ANOVA) asserted for significant statistics differences (*p* < 0.001) within species, membrane types, as well as differences within the interaction (species and membrane). The effect size complemented the statistical null hypothesis test and showed a potential real significance of the effect of the membrane in the assay (η^2^_p_ = 0.84), followed by the effect of the interaction between both factors (species and membrane) (η^2^_p_ = 0.64) and lastly, among species (η^2^_p_ = 0.44) (Table 5).

The RPE inhibited the growth of five of the six bacterial strains tested for the determination of the Minimum Inhibitory Concentration (MIC). The susceptibility of the Gram-positive bacteria toward RPE varied among the strains tested (MIC values from 8.51 to 68.12 µg/mL). The strongest antibacterial activity was exhibited against *S*. *epidermidis* CCCD, with a low MIC of 8.51 µg/mL, followed by *S. aureus* ATCC (MIC of 34.06 µg/mL) and the clinical strain isolate of *S. aureus* (MIC of 68.12 µg/mL). 

Comparing the two methods used to evaluate the antibacterial activity of the RPE (membranes and crude hydroalcoholic extracts), there is an agreement that the tested concentrations of RPE were more effective against the Gram-positive bacteria.

Gelatin-form membranes with concentrations of 40 and 200 g of RPE/100 g showed antimicrobial activity against *S. aureus* [38] and 13 different extracts of propolis from Serbia presented higher antimicrobial activities against Gram-positive bacteria than for Gram-negative bacteria, with both the disc-diffusion and microdilution methods [39]. The reference strains tested by the authors were the same used in the present research, with similar behavior after treatment with RPE. The incorporation of 10% RPE in chitosan-coated polypropylene membrane also enhanced antibacterial activity against *Bacillus cereus, Escherichia coli, Listeria monocytogenes, Salmonella typhimurium* and *S. aureus* [40]. Akca and coworkers indicated that RPE was more effective in inhibiting the Gram-positive/ATCC (*S. aureus, S. mutans* and *Enterococcus faecalis*) than the Gram-negative/ATCC (*Lactobacillus acidophilus* and *Prevotella intermedia*) bacteria in their planktonic state but less effective in biofilm form [41].

Microcapsules loaded with the RPE, composed of guar gum, pectin, maltodextrin, carbapol, carboxy-methylcellulose, stearic acid and colloidal silicon dioxide, used in the present experiments, displayed antibacterial activity (range of 200–400 μg/mL) in an agar-diffusion assay against *S. aureus* ATCC 25293 and *P. aeruginosa* ATCC 27853 with a MIC range from 135.87 up to 271.74 μg/mL for *S. aureus* ATCC and from 271.74 up to 543.48 μg/mL for *P. aeruginosa* ATCC [42]. The red propolis from Pernambuco (Northeast of Brazil) obtained during rainy and dry seasons presented different MIC values between them, against *P. aeruginosa* and a multi-resistant *S. aureus* [8]. Moreover, previous studies indicated different MIC ranges for chloroform and acetone extracts of the Brazilian red propolis to crude ethanol extract against *S. aureus, P. aeruginosa* and *Klebsiella pneumoniae* [27,43].

### 2.8. Antiallergic/Allergenic Potential

Stimulation of the skin by bioactive compounds can activate the response of keratinocytes and fibroblasts, which contributes to the recruitment of mast cells to the inflamed site, exacerbating an allergic reaction [44]. Mast cells are cells of the immune system frequently used to monitor the antiallergic/allergenic potential of bioactive compounds [18]. The activation of IgE receptors, FcεRI, on the membrane of mast cells represents the major event for the initiation of symptoms involved in allergic reactions [45]. The recognition of the antigen by the immune-complex FcεRI-IgE, on the mast cell membrane, triggers a cascade of intracellular reactions that precede the degranulation and consequent release of preformed allergic mediators such as histamine [46], serotonin [47] and beta-hexosaminidase [48], the latter used as a biological marker of degranulation [49]. Therefore, inhibition of antigen-stimulated mast cell degranulation represents an important approach to monitor the antiallergic/allergenic behavior of bioactive compounds. In this aspect, the rat basophilic leukemia (RBL-2H3) cells are generally used as a suitable model to study mast cell degranulation [50,51] since these cells are mucosal mast cells with comparable functions to primary mast cells [52]. Pretreatment of IgE-sensitized RBL-2H3 cells with RPE, followed by cell stimulation with antigen (DNP-BSA 0.1 µg/mL), showed a clear inhibition of β-hexosaminidase release mostly at RPE concentrations of 50 and 100 µg/mL (44% and 55% β-hexosaminidase inhibition, respectively) (Figure 6A). It is important to mention that the inhibitory activity of RPE (55%) was more potent than that exhibited by the mast cell stabilizer ketotifen fumarate (17% inhibition at a concentration of 100 ug mL^−1^), a commonly used positive control for mast cell degranulation assays [53]. Also, quercetin, a very well-known inhibitor of mast cell degranulation was used as a positive control, exhibiting 80% inhibition at a concentration of 30 µg/mL (Figure 6A) [54]. To avoid false-positive results about mast cell degranulation inhibition, the β-hexosaminidase catalytic activity (conversion of the substrate MUG to methylumbelliferone) was monitored in different concentrations of RPE. Figure 6B shows that the highest concentrations of RPE moderately inhibits β-hexosaminidase activity since 100 µg/mL of RPE causes 20% inhibition of enzyme activity, while the same concentration causes 55% inhibition of mast cell degranulation (Figure 6B). Consequently, the results have proven that the decrease in the percentage of the released β-hexosaminidase enzyme is mainly attributed to the ability of RPE to inhibit mast cell degranulation, denoting that the RPE indeed presents an important antiallergic potential. It is essential to highlight that RPE does not stimulate mast cell degranulation, which suggests that RPE does not present allergenic potential.

## 3. Discussion

In recent years, the development of products for wound healing evoked special interest. Polymeric materials have been widely used in the production of dressings, due to their biocompatibility, biodegradability, non-toxicity, cost-benefit and possibility of association with bioactive natural products [55]. Currently, there is a large number of curative products based on carboxymethylcellulose (CMC) and associations with other polymers described in the literature. However, NaCMC is less used for the development of products for wound healing [56]. In this study, topical membrane formulations based on NaCMC, loaded with red propolis extract (RPE), were developed. 

The RPE showed a phytochemical profile containing polyphenols such as phenolic acids, flavan-3-ols (catechins), flavonols, chalcones, isoflavones, isoflavans, pterocarpanes and biflavonoids [57,58,59] terpenes, homopterocarpin and benzophenones. Studies in the literature show that propolis produced in a tropical region has a great variety of chemical compounds, being strongly related to the botanical origin, flora, seasonal variation and the extraction technique used [50]. LC-Orbitrap-FTMS allowed the identification of common markers of red propolis in northeastern Brazil such as formononetin, daidzein, liquiritigenin, biochanin A, isoformononetin (Table 1), consistent with other studies [57,58,59]. Also, compounds such as medicarpin, guttiferones A, B and C, retusapurpurin A and B, xanthokimol, vestitol and neo-vestiol (Table 1) were identified in the extract, in lesser quantities and have important biologic activities [58]. Medicarpin is an important compound of the homopterocarpin class and is described as a potent cytotoxic compound on tumor lineage cells [59]. The benzophenones identified in these studies are reported as substances, which have antitumor, antibacterial, plasmodicidal, anti-HIV and other activities [58]. Propolis has been widely used and studied in different areas of medicine in various countries. The biologic potential of red propolis is directly related to its chemical composition. The antimicrobial and antioxidant activity of Brazilian red propolis is widely known in the literature. The membranes loaded with the RPE showed high antioxidant activity (IC_50_ ≥ 80.3 ± 1.7) as a function of the percentage of phenolic content released (Table 3).

The thermograms of hydrogel-based polymeric membranes (Figure 2A–D) show multiple stages of degradation. The use of citric acid (CA) as a crosslinking agent has a considerable influence on the flexibility and resistance to the handling of the membranes. In previous studies, citric acid was used to crosslink bioplastic agarose, where positive effects of this type of chemical crosslinking on the resistance, swelling, degradation and thermal stability of the biopolymer were observed. According to the present study, the tensile strength of the biopolymer doubled as compared to its non-crosslinked form [60]. A significant improvement in the resistance, hydrophilicity and general rheological properties of the hydrogels with citric acid was also observed by in vitro tests, comparing the hydrogel before and after the crosslinking [61].

The membranes developed in this work are cost-effective and easy to produce, using only NaCMC as the polymeric matrix, with proven antioxidant and microbiological activity. Several membranes or films are developed through the association of two or more polymers, increasing the production steps and costs. Ospanova et al. [62] produced films with the association of chitosan and NaCMC polymers in triclosan, chlorhexidine, silver ions and iodine plates for application as medical and biological implants due to their antibacterial potential. The results obtained showed good antimicrobial activity of the films [62]. In another study, De Lima et al. developed and evaluated the in vivo behavior of PVP/CMC/Agar hydrogel membranes loaded with silver nanoparticles for applications in wound healing. The results obtained revealed an intense formation of re-epithelization and neovascularization caused by the increased swelling capacity of the hydrogel membrane based on CMC [63]. However, Wong et al. [2] developed films using only NaCMC as a polymeric matrix and evaluated the antimicrobial activity and its application for wound healing. The developed NaCMC films showed efficacy in removing microbes from the wound bed by attaching a microbe to the polymer and accelerating healing [2].

The phenolic compounds present in RPE can cause functional and structural damage to the bacterial cytoplasmic membrane or cell wall [27]. The high performance of propolis-loaded membranes against bacteria is in line with other findings, using propolis extracts or other membrane formulations. Also, propolis has a broad-spectrum with activities against Gram-positive and Gram-negative bacteria, yeasts and filamentous fungi associated with varying degrees of pathogenicity in humans [40,64,65]. Interestingly, the present data showed that the *S. aureus* strain obtained directly from a hospitalized patient (Clinical/SAC) was less inhibited after contact with the M3 compared to the reference strains of *S. aureus* (SAP) and *S. epidermidis* (SEP), both Gram-positive bacteria. The same pattern was observed in the MIC determination, where *S. aureus* and *P. aeruginosa* from patients were less susceptible, with higher MIC values, in comparison to ATCC strains. The complete genomic analyses of *S. aureus* ATCC 25923 demonstrated that this strain is sensitive to a variety of antibiotics, including methicillin [66] and similarly, *S. aureus* (SAC) had similar behavior in our tests with M3. This clinical *S. aureus* strain may have some resistance mechanism decreasing its susceptibility to propolis; however, additional tests are necessary to confirm this hypothesis. 

Because citric acid (CA) is a useful preservative agent, it was used as a crosslinker agent (2×) in the M3 and it appears that this crosslinker may have enhanced the RPE antibacterial action of M3 on Gram-positive bacteria. Similarly, synergistic interactions of nisin-peptide and CA affected antibacterial activity against *S. aureus* and *L. monocytogenes* [65]. CA solution also showed antibacterial effects on facultative and obligate anaerobes [67]. Additionally, regarding the membrane distension observed at 24 h, the humidity of the culture medium interacting with each membrane tested (M/C-type) may have contributed to this distension. The same pattern can also be expected in skin wounds, once their initial exudate ensures a humid environment [68] and a moisture balance can be sought during wound treatment with sterile saline to promote distention and larger membrane coverage, in a potential treatment under wounds.

The pharmacologic properties of propolis are not limited to its antimicrobial activity. As shown in this work, Brazilian RPE inhibits mast cell degranulation induced by antigens, which corroborates the results obtained by other authors [17,20]. According to Nakamura and collaborators (2010), the effective doses to inhibit the allergic process diverge depending on the region where the propolis was collected. The chemical composition of propolis changes considerably according to the season and geographic location, which leads to variations in the biologic properties observed for different propolis extracts [16,24,69] For example, Brazilian propolis and Chinese propolis differ in their abilities to inhibit the release of histamine from rat peritoneal mast cells induced by compound 48/80 or concanavalin A [20], a behavior attributed to their chemical composition. Kaempferol and chrysin, both flavonoids, were identified as the main antiallergic components in ethanol extracts of Chinese propolis [17,70]. In the present study, both flavonoids were also identified in the Brazilian RPE, suggesting these substances may be responsible for the antiallergic properties of the extract. Despite the broad biologic applicability of propolis from different parts of the world, there have been reported cases of allergies to propolis [24,71]. Because the chemical composition of propolis can vary, some types of propolis can cause allergic reactions, while others do not [23]. In this context, it is important to highlight that Brazilian red propolis does not trigger mast cell stimulation responsible for allergic diseases, suggesting that the extract does not have allergenic properties, a matter of great importance for the wound healing process and other pharmacologic properties. Consequently, the RPE incorporated into the present biomembrane can be safely used to treat wounds, broad-spectrum bacterial infections and bacterial inflammatory processes. 

## 4. Materials and Methods 

### 4.1. Materials

Sodium carboxymethylcellulose (NaCMC) 3000 (Denver Especialidades Químicas, São Paulo, Brazil), citric acid (CA) (Êxodo Científica, São Paulo, Brazil) and propylene glycol (Synth, São Paulo, Brazil) were used in the production of the hydrogel membranes. The Folin-Ciocalteu reagent (Êxodo Científica, São Paulo, Brazil), 2,2-diphenyl-1-picrylhydrazyl radical (DPPH), gallic acid, quercetin, Trolox and 2,4,6-Tris(2-76 pyridyl)-s-triazine were purchased from Sigma Aldrich (Steinheim, Germany). The flavonoids, namely, chrysin, catechin, pinocembrin, kaempferol, daidzein, genistein, naringenin, galangin, formononetin, biochanin A, catechin, caffeic acid, ferulic acid and p-coumaric acid, were acquired from Sigma-Aldrich (St. Louis, MO, USA). Epicatechin and liquiritigenin were acquired from Extrasynthese^®^ (Lyon Nord, France) and were used as analytical standards. High performance liquid chromatography (HPLC) grade methanol was purchased from J.T. Baker (Mallinckrodt, Mexico) and acetonitrile was purchased from Fisher Scientific (Leicestershire, UK) and the Milli-Q grade water was produced in a lab. Sodium carbonate was supplied by Vetec Quíımica Fina (Rio de Janeiro, Brazil). Anhydrous monobasic sodium phosphate (Neon Commercial, São Paulo, Brazil) was used in the phenol release test. Brain heart infusion (BHI) broth (BD), nutrient agar Merck (Darmstadt, Germany), *Staphylococcus aureus* (ATCC 25923), *Staphylococcus epidermidis* (CCCD 5010) and *Pseudomonas aeruginosa* (ATCC 27853), besides two clinical isolates (*S. aureus and P. aeruginosa*), 2,3,5-triphenylltetrazolium chloride (TTC), polysorbate 80 (Êxodo Científica, São Paulo, Brazil) were used in the evaluation of the antimicrobial activity. All reagents were of analytical grade and the stock solutions and buffers were prepared with Milli-Q purified water. 4-Methylumbelliferyl-*N*-acetyl-b-d-glucosaminide (MUG), ketotifen fumarate, 3-(4,5-dimethylthiazol-2-yl)-2,5-diphenyltetrazolium bromide (MTT), dimethyl sulfoxide (DMSO); and bovine serum albumin (BSA) from Gibco (Grand Island, NY, USA). BSA conjugated with an average of 15 dinitrophenyl groups (DNP-BSA). Mouse monoclonal anti-DNP IgE from concentrated cell culture was purified as reported [50].

### 4.2. Red Propolis Extract (RPE)

Crude red propolis (raw material) was collected in February 2016, from Marechal Deodoro-Alagoas, Brazil (S 9°42′10.2924 and W 35°554′21.5316). The raw propolis (300.0 g) was used to obtain the extract, using the maceration extraction-method, with 1000 mL of commercial ethyl alcohol (80° GL), at room temperature. The solvent exchange was performed twice every 48 h, with daily stirring in the maceration process. The extraction process was performed in the dark, due to the photosensitivity of the extract. The RPE was filtered and then packed in an amber glass bottle to guarantee the preservation of bioactive compounds. The RPE was submitted to evaporation of the solvent, using a rotary evaporator (Fisatom, São Paulo-Brasil) at 60–80 °C. A solid mass (150.0 g) was obtained and was stored in a freezer (−20 °C) until the membrane preparation and characterization analyses.

### 4.3. Biopolymeric Hydrogel Membrane Loaded With Red Propolis Extract (RPE)

To prepare the hydrogel membranes, the biopolymer sodium carboxymethylcellulose (NaCMC) was mixed with citric acid as the crosslinking agent. An amount (5.0 g) NaCMC powder was solubilized in 100 mL distilled water under mechanical stirring (IKA RW 20 digital), at 800 rpm for 25 min. Subsequently, a volume (20 mL) of an aqueous solution containing 0.5% citric acid was poured onto 10.0 g of the (NaCMC) to obtain the M1 and M2 membrane-hydrogels and to the M3 membrane-hydrogel, an aqueous solution of 1% citric acid was added. The contents were mixed and then placed in an ultrasound apparatus (Unique USC-800) for homogenization [31]. Previously, the soluble solids content of the rotary evaporated RPE was assessed. Exactly 1g of RPE was added to the aluminum plate of an infrared drying balance from Shimadzu (Tokyo, Japan) programmed in automatic mode to warm up to a temperature of 120 °C until a constant weight of the sample, determining the content of solids and moisture. Then, the red propolis extract was weighed on an analytical balance equivalent to 10% of the total mass of the formulation, solubilized with 1 mL of absolute ethanol and incorporated into the membrane hydrogel. The mixture was subjected to mechanical stirring (IKA RW 20 digital), at 800 rpm for 10 min. The M1, M2 and M3 were transferred to 7 cm diameter molds and placed in an air circulation oven at 37 °C to permit evaporation of the solvent for periods of 24 and 48 h to obtain the biopolymeric membranes. The same procedure was performed to obtain the P1, P2 and P3 without RPE. The whole biopolymer membranes were removed from the molds without rupture, using tweezers, wrapped in film paper and conditioned at room temperature.

### 4.4. Thermogravimetric Analysis (TGA)

The thermogravimetric curves of the M1, M2 and M3 and hydroalcoholic RPE were obtained on a Shimadzu model TGA-50H apparatus calibrated with calcium oxalate standard. A nitrogen atmosphere was used at a flow of 20 mL min^−1^. The temperature ranged from 25 to 900 °C, with a heating rate of 10 °C min^−1^. A platinum crucible was used and the mass of each sample was 5.0 mg ± 10%. The data were analyzed using Shimadzu’s Tasys software. 

### 4.5. Attenuated Total Reflectance Coupled to Fourier-Transform Infrared (ATR-FTIR)

The ATR-FTIR spectra for the RPE and membranes, in the range of 4000 to 700 cm^−1^, were obtained on a Nicolet iS 10 FTIR spectrophotometer from Thermo Fisher Scientific in the ATR mode with 64 scans and 4 cm^−1^ resolution. The spectra were obtained at room temperature (~20 °C), with direct addition of the membranes and RPE into the apparatus, without previous treatment. This assay was performed to identify the presence of chemical functional groups.

### 4.6. Phytochemical Analysis

#### 4.6.1. LC-Orbitrap-FTMS of Red Propolis Extract (RPE)

The chromatographic profile of the RPE was performed using liquid chromatography, with an ultra-performance HPLC coupled to a diode array detector (UPLC-DAD) from Shimadzu (Tokyo, Japan). The propolis tincture was prepared at 100 mg/mL in ethanol and diluted to a concentration of 1 mg/mL and used in LC-Orbitrap-FTMS. The LC-orbitrap-FTMS from Thermo Scientific was used, under the following conditions. The stationary phase was a C18 column from ACE (100 × 4.6 mm; 5 µm) and the flow rate was 0.30 mL/min. The mobile phase consisted of (A) 0.1% formic acid in water: 0.1% of formic acid in acetonitrile (B) (*v*:*v*). The column was eluted in gradient mode, as follows: starting with 30% of (B), increasing to 45% in 6 min, 60 in 10 min, 75 in 14 min, 90 in 18 min and 100 in 22 min and held at 100% B between 22–47 min; then, the gradient decreased to 30 of (B) in 52 min and held at 30% B between 52–58 min. The FTMS was set to acquire ions in negative mode, with a needle voltage of 4.0 kV and sheath gas and auxiliary gas flows of 50 and 10 arbitrary units. The instrument was scanned over the range of 50 to 1200 amu. A volume of 10 μL was injected into the LC-orbitrap-FTMS. The spectra were acquired in negative ion mode with the same source settings as described above and with a collision energy of 35 V. Mzmine 2.32 version software was used to check the raw LC–Orbitrap-FTMS data and generate the MS based chromatograms which were deconvoluted, deisotoped, normalized, adduct and complex removed and masses and formula of the major chromatographic peaks. Mzmine software from each chromatographic peak was selected based on the peak area and putatively identified by searching for the accurate mass in Dictionary of Natural Products (version 2013), then identified by online library connected to the PubChem database.

#### 4.6.2. Total Polyphenolic Content (TPC) for RPE and Membranes

The Folin-Ciocalteu (FC) method was used for total phenolic contents (TPC) of membranes [72]. M1, M2 and M3 were weighed and corresponded to a concentration of 25 µg/mL of RPE incorporated in the polymer matrix. The reaction system consisted of adding 120 µL of ethanolic solutions of M1, M2 and M3, 180 µL of distilled water, 300 µL of FC reagent and after 2 min, 2400 µL of a 5 % (*w*/*v*) sodium carbonate solution. The reaction was shaken in the dark for 20 min. for the oxidation of phenols. The flask was then rapidly cooled and the developed color was read at 760 nm in a UV-Vis spectrophotometer (Agilent 8453, Agilent Technologies, Santa Clara, CA, USA). A change in color from a greenish to a blue color was observed. To calculate the percentage of total phenols, present in the sample a calibration curve constructed with gallic acid (GA) in ethanol (0.7–7.0 mg/mL) was used. The TPC was expressed as mg of GA equivalents per gram/dry weight of propolis extract.

#### 4.6.3. Release of Phenolic Compounds

M1, M2 and M3 were cut into discs (diameter = 20 mm), weighed (0.015 g) on an analytical balance (Shimadzu AUY220. Then the membrane discs were individually immersed in 30 mL of phosphate buffer solution (pH = 6.86) and corresponded to a concentration of 50 µg/mL of RPE incorporated in the polymer matrix. The samples were kept in an oven (Cienlab) at 36 °C and, after 1, 6, 24 and 48 h, an aliquot was collected of 0.5 mL of the buffer solution for quantification of total phenolic compounds by the Folin Ciocalteau method [29,72].

#### 4.6.4. DPPH Assay for RPE and Membranes

Radical-scavenging antioxidant activity of RPE and the membranes (M1, M2 and M3) was determined by the DPPH assay, according to the methods described in the literature [72,73] with a few modifications. The membranes were monitored by measuring the decrease in absorbance of the solutions, at different concentrations and absolute ethanol was used as a negative control.

The reaction system consisted of adding 0.30 mL of samples in ethanol (5–25 μg/mL) and 2.70 mL of DPPH^●^ solution (40 μg/mL in methanol). The mixture was homogenized and stored in the dark for 30 min. and the measurements were performed at 516 nm using a UV-vis spectrophotometer (Agilent 8453, Santa Clara, CA, USA). The percentage of inhibition or the IC_50_ (half maximal inhibitory concentration) was calculated graphically, using a calibration curve, in the linear range, by plotting the extract concentration versus the corresponding scavenging effect (I%, inhibition percentage), at 30 min. The value of I% was calculated using the Equation (1): I% = [(A_0_ − A_1_)/A_0_] × 100,(1)
where A_0_ is the absorbance of the control and A_1_ the absorbance in the presence of the extract.

#### 4.6.5. FRAP Assay

The ferric reducing ability was performed according to the method described in the literature [72]. M1, M2 and M3 were weighed and corresponded to a concentration of 25 µg/mL of RPE incorporated in the polymer matrix. FRAP reagent was prepared by addition 2.5 mL of a solution of TPTZ (10 mM) in HCl (40 mM), 2.5 mL of FeCl_3_ (20 mM) and 25 mL of 300 mM acetate buffer (pH 3.6). Sample aliquots of 90 μL were mixed with 270 μL of distilled water and 2.7 mL of FRAP reagent and incubated at 37 °C for 30 min. The absorbance of the reaction mixture was measured, at 595 nm and a calibration curve was prepared with Trolox^®^ (0.04 to 7.50 μg/mL). The analyzes were performed in triplicate and the results are expressed as Trolox Equivalent Antioxidant Capacities (TEAC), in mg of Trolox/g of dry matter content.

### 4.7. Antimicrobial Activity of RPE and Membranes

Susceptibility tests were performed using the reference strains of *Staphylococcus aureus* ATCC 25923, *Staphylococcus epidermidis* CCCD 5010 and *Pseudomonas aeruginosa* ATCC 27853, along with two clinical isolates of *S. aureus* and *P. aeruginosa*. Bacterial cultures were prepared in agar Brain-Heart Infusion (BHI) broth, according to the manufacturer’s instructions and incubated for 24 h, at 37 °C. Suspensions of bacteria were adjusted to 0.5 McFarland standard turbidity in sterile saline (0.9% NaCl), which corresponded to 10^7^–10^8^ CFU/mL.

The Kirby-Bauer disk diffusion method [74] was performed for the initial screening of antibacterial activity of M1, M2 and M3 and their controls without red propolis (P1, P2 and P3), according to the Clinical and Laboratory Standard Institute (CLSI, 2017), with modifications. Experiments were performed in Class II (Type A2) biologic safety cabinet and the membranes were exposed to ultraviolet (UV) light for 30 min, for effective sterilization. Petri dishes with Müller-Hinton agar (MHA) solid medium were prepared and the disc-diffusion assay was employed by adding each membrane (tests and controls) on the surface of MHA culture plates, previously inoculated with the adjusted suspensions (10^7^–10^8^ CFU /mL) of the reference and clinical microorganisms separately (*S. aureus, P. aeruginosa* and *S. epidermidis*), in triplicate. The inoculated Petri dishes were incubated for 48 h at 37 °C and the results were evaluated after 24 h, by measuring the inhibition zone in millimeters (mm) around each membrane, subtracting 10 mm (initial size) and respective distensions.

To evaluate different concentrations of RPE (M1, M2 and M3), the minimum inhibitory concentration (MIC) of these extracts was determined by the broth microdilution method in 96-well microplates with 0.1 mL/well of the RPMI-1640 medium, following the CLSI instructions. The RPE was prepared at a concentration of 2 mg/mL in a solution of distilled water/polysorbate 80 (80:20, *v*/*v*). The dilutions of 0.1 mL of RPE were suspended in serial concentrations: 1090 µg /mL, 545, 272.5, 136.25, 68.12, 34.06, 17.03, 8.51, 4.25, 2.12 µg /mL and 0 (negative control), plus a sterility control. The microorganism inocula (0.01 mL of each) were added to each well and the plates were incubated for 24 h at 37 °C. All dilutions and bacterial strains were tested, in triplicate, in the same microplate. After the incubation period, 0.02 mL of 2,3,5-triphenylltetrazolium chloride (TTC) was added to each well, as an indicator of cell viability with 1 h of incubation at 37 °C (only viable cells are stained by TTC). MIC-values were determined as the lowest concentration of RPE, at which no visible growth occurred, without TTC-staining.

### 4.8. Antiallergic/Allergenic Potential

#### 4.8.1. Mast Cell Culture

Rat basophilic leukemia (RBL-2H3) cells [46] were maintained in monolayer culture in minimum essential medium with *L*-glutamine and Eagle’s salt, supplemented with 20% fetal bovine serum (Atlanta Biological, Atlanta, GA., USA) and 50 μg/mL gentamicin sulfate (Invitrogen Corp., Carlsbad, CA, USA). The cells were used from 3 to 5 days after passage and released from culture flasks, by treatment with 0.5% trypsin-EDTA for 5 min. at 37 °C, centrifuged at 1000 rpm for 5 min, (centrifuge Sorvall Legend, model Mach 1.6 R. Thermo Fisher Scientific, Walthman, MA, USA) and resuspended at 1 × 10^6^ cell/mL. Reagents used were from Gibco (Carlsbad, CA, USA), except where indicated.

#### 4.8.2. β-Hexosaminidase Release Assay

IgE-mediated mast cell degranulation was conducted as described by Naal and others, with some modifications [50]. RBL-2H3 cells were initially sensitized with 1 μg/mL anti-DNP IgE (purified as previously described [75], plated onto 96-well plate at 5 × 10^5^ cell/mL for 24 h, at 37 °C with 5% CO_2_ atmosphere. Next, cells were washed twice with 200 μL of Tyrode’s buffer (pH 7.4) and pretreated, for 20 min, with different concentrations of RPE (2–100 μg/mL, diluted in Tyrode’s buffer from a DMSO stock solution), quercetin (30 μg/mL^−1^) and ketotifen fumarate (85 μg/mL). Cells were stimulated by the addition of 100 μL of Bovine Serum Albumin (BSA), 2,4-Dinitrophenylated (DNP-BSA) (final concentration 0.1 µg/mL) and incubated for 1 h at 37 °C, followed by cooling in an ice bath to stop degranulation. Cell degranulation was measured by determining the amount of released β-hexosaminidase. Therefore, 25 μL of cell supernatant and 100 μL of 1.2 mM β-hexosaminidase substrate (4-methylumbelliferyl-*N*-acetyl-β-d-glucosaminide, MUG, Sigma-Aldrich), in 0.05 M sodium acetate buffer (pH 4.4), were mixed in separate 96-well plates and incubated for 30 min, at 37 °C. The enzyme-substrate reaction was stopped by the addition of 175 μL of 0.1 mol/L carbonate buffer, pH 10. Controls without antigen were used to measure spontaneous release. The total β-hexosaminidase release was obtained by lysing cells with 0.1% Triton-X 100 before removing the supernatant. Released β-hexosaminidase was quantitated by measuring the fluorescence intensity of the product of β-hexosaminidase mediated cleavage of MUG, methylumbelliferone, in a microplate reader (BioTEK, Winooski, VT, USA) using 360 nm excitation and 450 nm emission filters. The percentage of stimulated β-hexosaminidase-release was calculated, according to Equation (2). The inhibitory effect of RPE on β-hexosaminidase release was compared to quercetin and ketotifen fumarate. Results are expressed as mean ± SD. Statistical significance was assessed by one-way ANOVA followed by Dunnett’s post-hoc pairwise comparisons. Differences were considered statistically significant at * *p* < 0.05, ** *p* < 0.01 and *** *p* < 0.001.
(2)$$\%\beta -\mathrm{Hex}=\frac{S-N}{T-N}\times 100.$$

*N*: Normal, fluorescence from the vehicle (Tyrode’s buffer). *S*: Sample, fluorescence from (+) DNP-BSA, (+) or (−) red propolis samples. *T*: Total, fluorescence from Triton X-100-lysed cell samples.

#### 4.8.3. β-Hexosaminidase Activity Assay

RBL-2H3 cell suspensions, at 5 × 10^5^ cell/mL, were sonicated for 20 min for cell lysis and release of β-hexosaminidase enzyme. After centrifugation at 1000× *g* for 5 min, 45 μL of supernatant was incubated with 5 μL of RPE sample solutions (2–100 μg/mL) and 50 μL of β-hexosaminidase substrate (MUG) solution (final concentration 1.2 mM), for 60 min. The enzyme-substrate reaction was stopped by the addition of 200 μL of stop solution (0.1 M carbonate buffer, pH 10.0). β-Hexosaminidase activity was determined from fluorescence measurements of hydrolyzed MUG, as described above. Samples not treated with propolis were taken as 100% of β-hexosaminidase activity. Statistical significance was assessed, as described under 4.8.2.

### 4.9. Statistical Analysis

Statistical analysis of physicochemical and chemical data was carried out using GraphPad Prism 5.01 (GraphPad Software, San Diego, IL, USA). Analysis of variance and least significant difference tests were conducted to identify differences among the means, *p*-value < 0.05 was regarded as significant.

The data from the disk-diffusion assays were treated using the open-source and cross-platform software JASP (Version 0.8.3.1) (Amsterdam, The Netherlands) to evaluate the antibiotic activity of membranes against bacteria. Analyses of variance (ANOVA) of the halo diameter means (expressed in millimeters) was conducted using Volk-Selk maximum *p*-ratio and was based on the *p*-value, followed by the estimates of effect sizes by η^2^, partial η^2^ and ω^2^. Significant differences between the means of the species, the membrane and the interaction between both were established by Tukey’s test (*p* < 0.001). The post-hoc Tukey’s correction test was accomplished for multiple comparisons between species (PAC, PAC, SAC and SAP) as well as between membrane groups (M1/M2/M3/P1/P2/P3). The significance of the value of the dependent variables (membrane, incubation and the interaction membrane* incubation) was evaluated by the “tests of between-subjects effect” that was carried out to detect differences among mean values inhibition zone for each membrane. *p*-values of <0.001 were accepted as significant.

## 5. Conclusions

In this study, low-cost membranes were developed based on NaCMC loaded with RPE. The RPE-loaded membrane is very promising, once RPE is a rich source of phenolic compounds, containing a huge variety of flavonoids. The citric acid used in membrane formulations as a crosslinking agent influenced the thermal stability and the release profile of phenolic compounds, revealing greater stability for M3 and greater release of these compounds by M1. The RPE released from the membranes showed high antioxidant activity, mainly for M1. All membranes showed antibacterial activity against Gram-positive bacteria, with significantly higher performance for M3. The RPE showed inhibitory activity against the secretion of allergic mediators released from antigen-stimulated mast cells, suggesting that the extract has no allergenic properties. The physical barrier formed by this material and the action of RPE as an antioxidant and antibacterial against the main bacteria involved in skin infections can be determining factors in the healing process. 

## Figures and Tables

**Figure 1 molecules-25-05811-f001:**
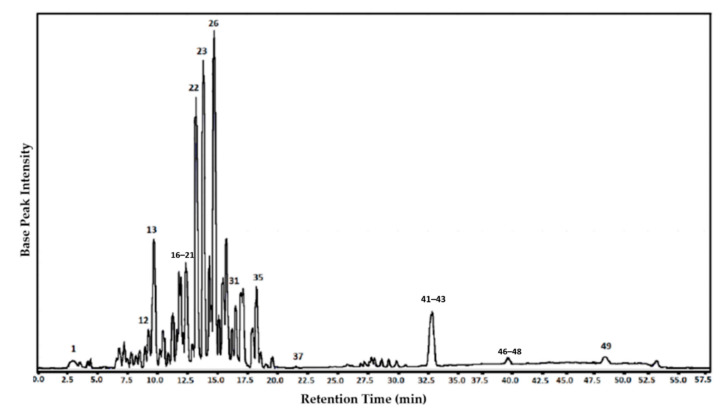
Chromatogram of red propolis extract (RPE) obtained using Orbitrap-coupled liquid chromatography and Fourier transform mass spectrometry (LC-Orbitrap-FTMS).

**Figure 2 molecules-25-05811-f002:**
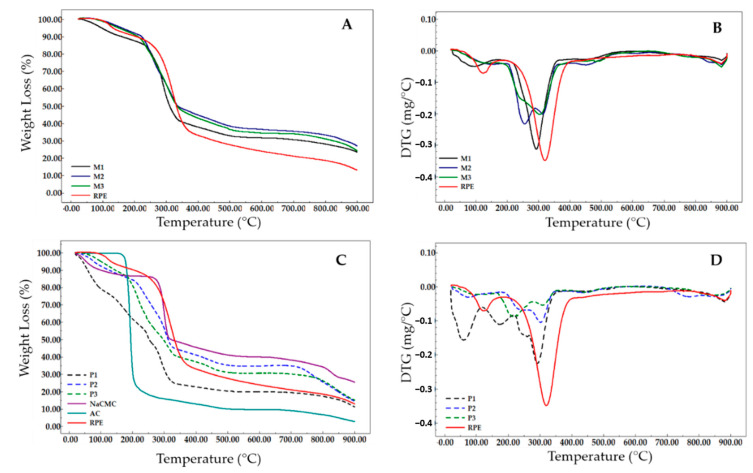
Thermogravimetric (**A**) and derivative thermogravimetric (**B**) curves of polymeric hydrogel membranes loaded with red propolis extract (RPE). Thermogravimetric (**C**) and thermogravimetric derivative (**D**) curves of polymeric hydrogel membranes of the controls (P1, P2 and P3), RPE, sodium carboxymethylcellulose (NaCMC) and, citric acid (CA).

**Figure 3 molecules-25-05811-f003:**
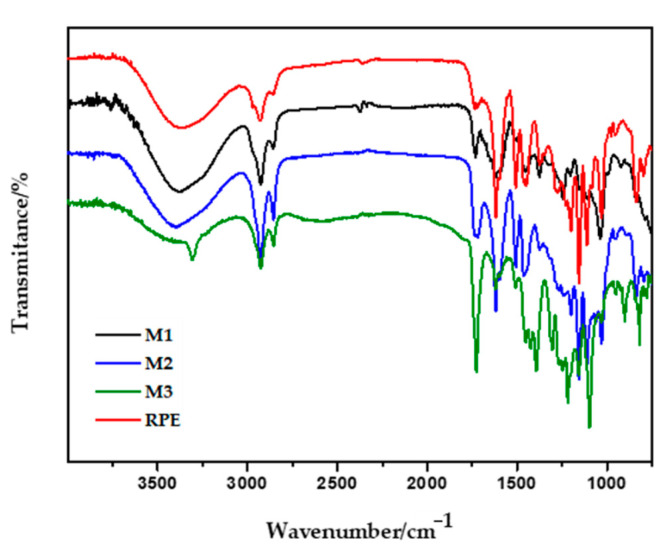
Attenuated total reflectance-Fourier transform infrared spectroscopy (ATR-FTIR) related to the polymer membranes loaded with the red propolis extract (RPE) (M1, M2, M3).

**Figure 4 molecules-25-05811-f004:**
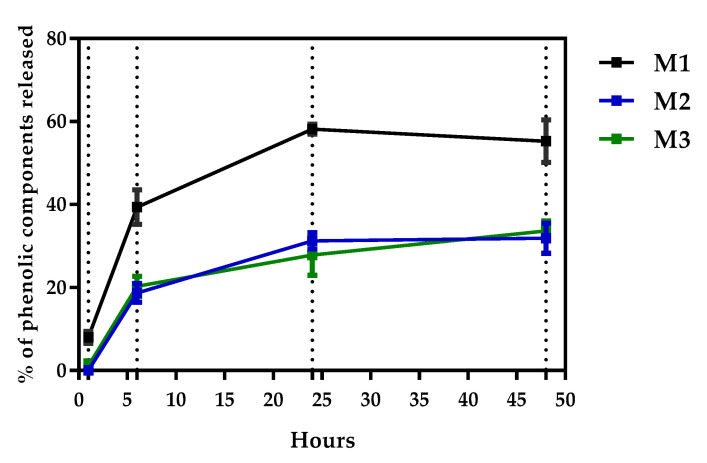
Percentage of the release of the phenolic components of the RPE trapped in the hydrogel matrix of M1, M2 and M3.

**Figure 5 molecules-25-05811-f005:**
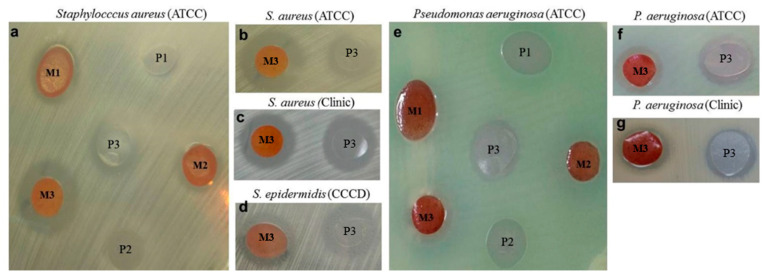
Antimicrobial activity of the membranes against Gram-positive and Gram-negative bacteria after 24 h by the disk-diffusion technique, using M1, M2 and M3 (loaded with RPE) and P1, P2 and P3 as their respective control membranes. (**a**) Zones of inhibition produced by the polymer hydrogel membranes loaded with RPE against *Staphylococcus aureus* ATCC 25923. The M3 against (**b**) *Staphylococcus aureus* ATCC 25923; (**c**) clinical isolated of *S. aureus;* (**d**) *S. epidermidis* CCCD 5010 and (**e**) Zone of inhibition produced by the polymer hydrogel membranes loaded with RPE against *Pseudomonas aeruginosa* ATCC 27853. The M3 against (**f**) *P. aeruginosa* ATCC 27853 and (**g**) clinical isolated of *P. aeruginosa.*

**Figure 6 molecules-25-05811-f006:**
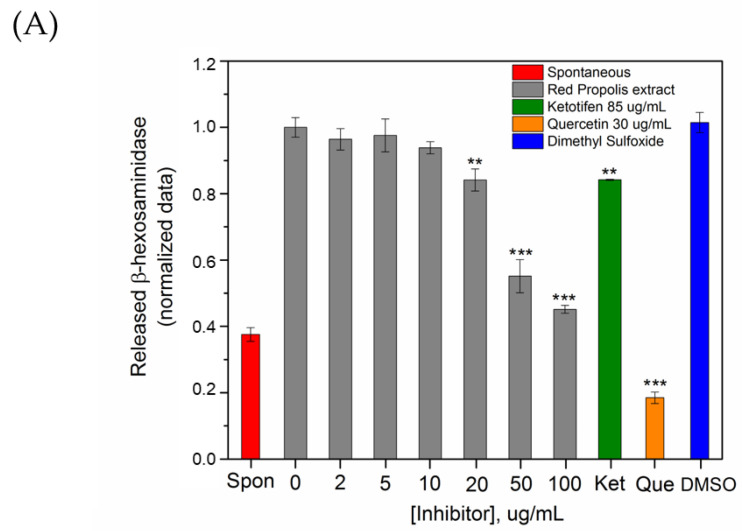
(**A**) Effect of red propolis extract (RPE), quercetin and ketotifen fumarate on the release of the beta-hexosaminidase enzyme, from mast cells stimulated by antigen. Anti-DNP IgE-sensitized RBL-2H3 cells were treated with red propolis extract, quercetin or ketotifen fumarate at the indicated concentrations, followed by antigen (DNP-BSA) stimulation and addition of the substrate metilumbeliferil-N-acetil-β-D-glucosaminida (MUG). Released β-hexosaminidase was determined as described in the materials and methods section. Data are expressed as mean ± SD of three independent experiments. (**B**) Effect of RPE on the β-hexosaminidase activity. Enzyme activity was quantitated by fluorescence measurements of methyl-umbelliferone after cell supernatants were incubated with RPE (1–100 μg mL^−1^) and 1.2 mmol L^−1^ MUG solution. One-way ANOVA, followed by Dunnett’s test, used for both assay indicates statistical significance of * *p* < 0.05, ** *p* < 0.01 and *** *p* < 0.001 in comparison to the placebo samples.

**Table 1 molecules-25-05811-t001:** Identification and confirmation of some markers of the red propolis extract (RPE) using Orbitrap-coupled liquid chromatography and Fourier transform mass spectrometry (LC-Orbitrap-FTMS).

Peak	RT (min)	[M − H]^−^ (*m*/*z*)	MW	Formulae	Compound
1	2.95	179.0556	180.16	C_9_H_8_O_4_	Caffeic acid
2	2.98	193.0502	194.18	C_10_H_10_O_4_	Ferulic acid
3	3.00	178.0556	179.05	C_9_H_8_O_4_	Umbellic acid
4	3.04	163.0243	164.16	C_9_H_8_O_3_	*p*-Coumaric acid
5	3.10	475.1232	476.43	C_23_H_24_O_11_	7-*O*-beta-glucopyranosyl-4′-hydroxy-5-methoxyisoflavone
6	4.50	461.1023	462.40	C_22_H_22_O_11_	6-Methoxyluteolin 7-rhamnoside
7	7.05	269.0811	270.24	C_15_H_10_O_5_	Genistein
8	7.35	285.0395	286.24	C_15_H_10_O_5_	Kaempferol
9	8.04	289.0711	290.27	C_15_H_14_O_6_	Catechin
10	8.28	287.0553	288.25	C_15_H_12_O_6_	Dalbergioidin
11	8.83	289.0711	290.27	C_15_H_14_O_6_	Epicatechin
12	8.95	253.0499	254.24	C_15_H_10_O_4_	Daidzein
13	9.70	255.0654	256.27	C_15_H_12_O_4_	Liquiritigenin
14	10.5	283.0384	284.26	C_16_H_12_O_5_	2′-Hydroxyformononetin
15	11.3	331.0810	332.30	C_17_H_16_O_7_	Evernic acid
16–17	11.9	271.0602	272.25	C_15_H_12_O_5_	Narigenin/Pinobanksin
18	12.4	285.0758	286.24	C_15_H_10_O_6_	Calycosin
19	12.8	521.1600	522.173	C_32_H_26_O_7_	Retusapurpurin B
20	13.2	521.1600	522.173	C_32_H_26_O_7_	Retusapurpurin A
21	13.4	255.0654	256.27	C_15_H_12_O_4_	Isoliquiritigenin
22–23	13.77	267.0655	268.28	C_16_H_12_O_4_	Formononetin / Isoformononetin
24	15.5	253.087	254.28	C_16_H_14_O_3_	6-Methoxyflavanone
25	15.5	287.056	288.25	C_15_H_12_O_6_	6-Hydroxynaringenin
26	14.2	269.0812	270.28	C_16_H_14_O_4_	4,4′-dihydroxy-2-methoxychalcone
27	14.2	269.0812	270.32	C_16_H_14_O_4_	(7*S*)-Dalbergiphenol
28	14.66	271.0603	272.29	C_16_H_16_O_4_	Vestitol
29	15.10	269.0813	270.28	C_16_H_14_O_4_	Pinostrobin
30	15.10	269.0813	270.27	C_16_H_14_O_4_	Medicarpin
31	16.2	271.0607	272.29	C_16_H_16_O_4_	2′,6′-dihydroxy-4′-methoxydihydrochalcone
32	16.2	283.0657	284.26	C_16_H_12_O_5_	Thevetiaflavone
33	16.42	283.0603	284.26	C_16_H_12_O_5_	Biochanin A
34	16.73	253.0865	254.25	C_15_H_10_O_4_	Chrysin
35	16.87	255.1019	256.27	C_15_H_12_O_4_	Pinocembrin
36	17.0	539.1699	540.56	C_32_H_28_O_8_	3′,4′-di-*O*-benzyl-7-*O*-(2-hydroxyethyl)-3-*O*-methylquercetin
37	17.9	285.113	286.32	C_17_H_18_O_4_	Sativan
38	18.2	285.1131	286.32	C_17_H_18_O_4_	(3*S*)-7-*O*-methylvestitol
39	18.2	285.1131	286.32	C_17_H_18_O_4_	7,3′-Dihydroxy-4′-methoxy-8-methylflavan
40	21.4	425.1603	426.71	C_30_H_50_O	Cycloartenol/α-amyrin/β-amyrin
41	23.6	533.2906	534.69	C_33_H_42_O_6_	Hyperibone H
42	25.5	617.3480	618.82	C_38_H_50_O_7_	16-hydroxyguttiferone K
43	27.3	511.1383	512.50	C_30_H_24_O_8_	Rhuschalcone V
44	32.80	601.3533	602.80	C_38_H_50_O_6_	Guttiferone F
45	32.88	601.3533	602.80	C_38_H_50_O_6_	Xantochymol
46	32.90	601.3533	602.80	C_38_H_50_O_6_	Guttiferone E
47	34.10	347.2233	348.52	C_22_H_36_O_3_	Anacardic acid (6-pentadecylsalycilic acid)
48	39.24	669.4355	670.917	C_43_H_58_O_6_	Guttiferone C
49	39.24	669.4355	670.917	C_43_H_58_O_6_	Guttiferone D
50	39.24	669.4355	670.917	C_43_H_58_O_6_	Guttiferone B

**Table 2 molecules-25-05811-t002:** Compositions of membranes (M1, M2 and M3), loaded with red propolis extract (RPE).

Components	Composition (Proportion, *w*/*w*)					
	M1	M2	M3	P1	P2	P3
**Na** **CMC**	5.0	5.0	5.0	5.0	5.0	50
**Citric acid**	0.50	0.50	1.0	0.5	0.5	1.0
**Propylene glycol**	0.50	-	-	0.5	-	-
**RPE**	0.60	0.55	0.6	-	-	-
**Distilled water**	93.4	93.9	93.4	94.0	94.5	94.0

NaCMC = sodium carboxymethylcellulose; RPE = red propolis extract (10% of the solid components of the formulation); M = membrane; P = control.

**Table 3 molecules-25-05811-t003:** Total phenol content of fully solubilized membranes by the Folin-Ciocalteau method and antioxidant activity by the 2,2-diphenyl-1-picrylhydrazyl radical (DPPH) and Ferric Reducing Antioxidant Power (FRAP) ssays.

Samples	TPC (mg GAE * g^−1^)	DPPH^●^ μg/mL (IC_50_)	FRAP (mg TEAC ** g^−1^)
**M1**	167.1 ± 14.2 ^a^	83.3 ± 4.4 ^a^	57.3 ± 4.7 ^a^
**M2**	136.5 ± 3.7 ^b^	80.3 ± 1.7 ^a^	57.1 ± 1.8 ^a^
**M3**	134.5 ± 8.3 ^b^	87.0 ± 2.9 ^a^	58.5 ± 6.1 ^a^

TPC: Total phenol content; * Gallic Acid Equivalents (GAE). ** Trolox Equivalent Antioxidant Capacity (TEAC). Values are mean ± SD. Means with different letters (^a^ and ^b^) within a column are significantly different (*p* < 0.05).

**Table 4 molecules-25-05811-t004:** Antimicrobial activities of M1, M2 and M3 vs. P1, P2 and P3, using the disk diffusion technique.

Inhibition Halo (mm)						
Species	M1	P1	M2	P2	M3	P3
**SAP**	3.0	-	<1.0	-	7.0	1.0
**SAC**	2.0	-	>3.0	-	>4.0	>4.0
**SEP**	>2.0	1.0	>1.0	-	>5.0	>3.0
**PAP**	2.0	-	>1.0	-	>2.0	>2.0
**PAC**	-	-	-	-	>2.0	>2.0

Descriptive plot considering the interaction between species and membrane halos (*p* < 0.001). Values are mean ± SD from the assay in triplicate. M1 = membrane with RPE, sodium carboxymethylcellulose (NaCMC), without crosslinker agent (citric acid), M2 = with RPE, sodium carboxymethylcellulose (NaCMC) and 1× crosslinker agent (citric acid), M3 = with RPE, sodium carboxymethylcellulose (NaCMC) and 2× crosslinker agent (citric acid)] and P1, P2 and P3 (their controls, respectively) without RPE PAC = *Pseudomonas aeruginosa* (clinical), PAP = *Pseudomonas aeruginosa* (reference), SAC = *Staphylococcus aureus* (clinical), SAP = *Staphylococcus aureus* (reference) and SEP = *Staphylococcus epidermidis* (reference).

**Table 5 molecules-25-05811-t005:** Analysis of variance (ANOVA) of the antibacterial assay.

Experiments	Mean Square	F	*p*	VS-MPR *	η^2^	η^2^_p_	ω^2^
Species	7.76	11.77	<0.001	62312	0.09	0.44	0.08
Membrane	43.24	65.64	<0.001	1.475 × 10^20^	0.60	0.84	0.59
Species * Membrane	3.59	5.45	<0.001	151032	0.20	0.64	0.16
Residual	0.66						

VS-MPR: * Vovk-Sellke Maximum *p* -Ratio: based the *p*-value, the maximum possible odds in favor of H_1_ over H_0_ equals 1/(−e *p* log(*p*)) for *p* ≤ 0.37.

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
