# Peer review of "New Insights for Red Propolis of Alagoas—Chemical Constituents, Topical Membrane Formulations and Their Physicochemical and Biological Properties"

_molecules, 2020, doi:10.3390/molecules25245811_

Round 1
Reviewer 1 Report
I have gone through the manuscript titled " New Insights for Red Propolis of Alagoas: Chemical Constituents, Topical Membrane Formulations and Their Physicochemical and Biological Properties " It is well performed and presented as a study.
Abstract:
Other important analyses carried out in this manuscript should be included in the abstract.
Introduction:
The introduction is clear, but not covered the story well. The introduction should cover the literature on this topic as well. Also, recent references should be included.
Results:
The results are clear and important.
Discussion:
The discussion is somewhat well-written but still needs improvement.
The current data should be compared with previously published findings and how these new findings support the research question.
Materials and Methods:
please summarized it.
Conclusions:
please rephrase it
Line 57-74, please rephrase it
Line 178-195, please rephrase it
Line 325-354, please rephrase it
Manuscript need English editing.
Author Response
- We would like to thank reviewer1 for their excellent comments that allow for improving the manuscript.
- Abstract:
Other important analyses carried out in this manuscript should be included in the abstract.
Answer: Thanks. We have asked the editor to enlarge the abstract. It has now 260 words. Several results and methods have been added to the abstract,
The text is now:
The main objectives of this study were to evaluate the chemical constitution and allergenic potential of red propolis extract (RPE). They were evaluated, using HPLC and the release of β-hexosaminidase, respectively. A plethora of biologically active polyphenols and the absence of allergic responses were evinced. RPE inhibited the release of β-hexosaminidase, suggesting that the extract does not stimulate allergic responses. Additionally, the physicochemical properties and antibacterial activity of hydrogel membranes loaded with RPE were analyzed. Bio-polymeric hydrogel membranes (M) were obtained using 5% carboxymethylcellulose (M1 and M2), 1.0% citric acid (M3), and 10% RPE (for all). Their characterization was performed using thermal analysis, FTIR, total phenolic content, phenol release test and, antioxidant activity through DPPH and FRAP. The latter appointed to the similar antioxidant capacity of the M1, M2, and M3. The degradation profiles showed higher thermostability to M3, followed by M2 and M1. The incorporation of RPE into the matrices and the crosslinking of M3 were evinced by FTIR. There were differences in the release of phenolic compounds, with a higher release related to M1 and lower in the strongly crosslinked M3. The degradation profiles showed higher thermostability to M3, followed by M2 and M1. The antibacterial activity of the membranes was determined using the disc diffusion assay, in comparison with controls, obtained in the same way, without RPE. The membranes elicited antibacterial activity against Staphylococcus aureus and Staphylococcus epidermidis, with superior performance over M3. The hydrogel membranes loaded with RPE promote a physical barrier against bacterial skin infections and may be applied in the wound healing process.
- Introduction:
The introduction is clear, but not covered the story well. The introduction should cover the literature on this topic as well. Also, recent references should be included.
The introduction has been reformulated with the addition of more recent references. We improved the contextualization of polymeric NaCMC membranes in terms of cost-benefit and the importance of incorporating red propolis extract with relevant biological activities.
References 3-6, 15, 19, 56-57, 63-64 were included. They are highlighted in yellow.
- Results:
The results are clear and important. Line 178-195, please rephrase it. Thanks. The text is now:
The main functional groups of the membranes were investigated by ATR-FTIR. Figure 3 shows the ATR-FTIR spectra of the RPE and the RPE-incorporated membranes. For the RPE, the stretching of the bands OH at 3372 cm-1 representing the olic hydroxyl group was identified. The bands C–H3 (sp3) at 2932 cm-1 and, C–H2 (sp3) at 2856 cm-1 are peaks from the residual wax present in the hydroalcoholic extract. Bands at 1618, 1508, and 1448 cm-1 correspond to the nC=C of the aromatic ring and at 1030 cm-1 to the C-O bond (for flavonoids), as well as a band at 836 cm-1, relative to an angular deformation outside of the aromatic C-H plane, according to the literature [28, 31].
M1, M2, and M3 showed a similar profile in comparison with the RPE, proving the incorporation of the extract into the polymer matrix. The FTIR spectra of membranes revealed the simultaneous presence of sodium carboxylate (–COONa) and carboxylic acid (–COOH) in the NaCMC hydrogel, attributed respectively to bands in the region between 1622 - 1613 cm-1 and 1735 - 1729 cm-1 [32]. The reaction between citric acid and the hydroxyl groups of the NaCMC leads to crosslinking on the polymer matrix [33]. The crosslinking step through acidification of the hydrogel with citric acid (CA) caused a change in the spectrum, suggesting the substitution of Na+ for H+ in the chains of the NaCMC polymer. The reduction of the intensity of the -OH bands (3384 to 3466 cm-1) and the increase of the peak intensity in the region between 1735 to 1729 cm-1 of citric acid-crosslinked membranes were observed. It is suggested that the increase of citric acid concentration can cause the consumption of -OH due to intramolecular and intermolecular hydrogen bonds during the crosslinking reaction [34]. This consumption was observed mainly in the membranes without humectant (propylene glycol) and shown in M3, containing a double concentration of citric acid. This shows an increasing degree of crosslinking with a higher percentage of citric acid crosslinker added to the membrane. Depending on the increase in the concentration of CA, a decrease in the band representing sodium carboxylate (–COONa) and an increase in the band of carboxylic acid was observed (–COOH) [32].
- 7 - Line changes: 194 to 216
Discussion:
The discussion is somewhat well-written but still needs improvement.
We made a comparative contextualization of polymeric membranes based on NaCMC in terms of cost, ease of preparation,and effectiveness against microorganisms.
p.13 - Line changes: 388 to 396
Conclusions:
please rephrase it.
The conclusion was reformulated as requested. It is now:
In this study, low-cost membranes were developed based on NaCMC loaded with RPE. The RPE-loaded membrane is very promising, once RPE is a rich source of phenolic compounds, containing a huge variety of flavonoids. The citric acid used in membrane formulations as a crosslinking agent influenced the thermal stability and the release profile of phenolic compounds, revealing greater stability for M3 and greater release of these compounds by M1. The RPE released from the membranes showed high antioxidant activity, mainly for M1. All membranes showed antibacterial activity against Gram-positive bacteria, with significantly higher performance for M3. The RPE showed inhibitory activity against the secretion of allergic mediators released from antigen-stimulated mast cells, suggesting that the extract has no allergenic properties. The physical barrier formed by this material and the action of RPE as an antioxidant and antibacterial against the main bacteria involved in skin infections can be determining factors in the healing process.
Reviewer 2 Report
- Authors should include discussion as to why some of the control membranes (P3 specifically) showed antimicrobial activity in figure 5.
Author Response
Authors should include discussion as to why some of the control membranes (P3 specifically) showed antimicrobial activity in figure 5.
In the results
Otherwise, M3 promoted antibacterial activity against all bacteria, with a difference in the measurement of halo in comparison with P3 (citric acid 2x), which generated a smaller halo.
In the discussion:
Because citric acid (CA) is a useful preservative agent, it was used as a crosslinker agent (2x) in the M3, and it appears that this crosslinker may have enhanced the RPE antibacterial action of M3 on Gram-positive bacteria. Similarly, synergistic interactions of nisin-peptide and CA affected antibacterial activity against S. aureus and L. monocytogenes [66]. CA solution also showed antibacterial effects on facultative and obligate anaerobes [68].
(p. 9, line 280) in the Fig. 5 P. aeruginosa was altered to Pseudomonas aeruginosa;
- (p. 17, line 617) broth
Reviewer 3 Report
This article gathers together numerous experimental results regarding red propolis extract (RPE) from Brazil. After analyzing and describing the molecular composition (using a variety of instrumental methods) of RPE, the manuscript focuses primarily on the dermatologic properties, allergenic activity and anti-bacterial properties when RPE is added to hydrogel membranes used for wound healing.
It is very well written and from my reading, it requires only minor changes described below.
In general, authors should be consistent when referring to the bacteria that are mentioned by using italics throughout the manuscript.
In line 111, a period is missing after formulations
line 153 should be "occurs"
In Table 3 under DPPH studies, the units for concentration should be reported for IC50 values.
Author Response
Sorry and thanks. As requested, all microorganisms’ names in the subtopic 2.7 were adjusted to italic. (p.9 to 11; line 258 to 343)
- In line 111, a period is missing after formulations
- line 153 should be "occurs"
Thanks. Done. We added "occurred".
- In Table 3 under DPPH studies, the units for concentration should be reported for IC50 values.
Changes were made to lines 111 and 153 as requested. In addition, the μg / mL unit was added.
Round 2
Reviewer 1 Report
The authors have made changes to the manuscript, so I consider it can be accepted for publication.